# Synergistic Antibacterial Activity of Amorolfine Combined with Colistin Against *Acinetobacter baumannii*

**DOI:** 10.3390/ijms26073312

**Published:** 2025-04-02

**Authors:** Ting Lin, Shuaiyuan Liu, Xuan Chen, Fei Gao, Lu Liu, Daijie Chen, Yu Yin

**Affiliations:** 1School of Pharmacy, Shanghai Jiao Tong University, Shanghai 200240, China; lin-ting@sjtu.edu.cn (T.L.); lsy2023@sjtu.edu.cn (S.L.); belieforever97@sjtu.edu.cn (X.C.); feigao@sjtu.edu.cn (F.G.); liulu2000@sjtu.edu.cn (L.L.); cdj@sjtu.edu.cn (D.C.); 2State Key Laboratory of Microbial Metabolism, Shanghai Jiao Tong University, Shanghai 200240, China

**Keywords:** amorolfine, colistin, antibiotic adjuvant, *Acinetobacter baumannii*

## Abstract

Emerging resistance to colistin in *Acinetobacter baumannii* is concerning because of the limited therapeutic options for this important clinical pathogen. Given the shortage of new antibiotics, one strategy that has been proven to be therapeutically effective is to overcome antibiotic-resistant pathogens by combining existing antibiotics with another antibiotic or non-antibiotic. This study was designed to investigate the potential synergistic antibacterial activity of amorolfine, a morpholine antifungal drug, in combination with colistin against *A. baumannii*. In this work, antibiotic susceptibility testing, checkerboard assays, and time-kill curves were used to investigate the synergistic efficacy of colistin combined with amorolfine. The molecular mechanisms of combination therapy were analyzed using fluorometric assays, UV-vis spectroscopy, and molecular docking. Finally, we evaluated the in vivo efficacy of combination therapy against *A. baumannii*. In brief, the combination therapy showed significant synergistic activity against *A. baumannii* (FICI = 0.094). In addition, the combination of amorolfine improved the membrane disruption of colistin, and amorolfine exhibited the capacity of binding to DNA. Moreover, in a mouse sepsis model, this combination therapy increased survival compared to colistin monotherapy. Our findings demonstrated that amorolfine serves as a potential colistin adjuvant against *Acinetobacter baumannii*.

## 1. Introduction

*Acinetobacter baumannii* is a ubiquitous Gram-negative bacterium that is commonly isolated from the environment [1]. Its ability to form biofilm and its resistance to desiccation and disinfectants allows *A. baumannii* to thrive in the hospital environment and, as an opportunistic pathogen, has caused hospital-acquired infections [2,3]. In addition, due to its resistance to last-resort antibiotics, such as colistin, tigecycline, and carbapenems, *A. baumannii* has been classified as an ESKAPE pathogen by the WHO, and the research and development of new antibiotics is critically needed [4].

Colistin is a polypeptide antibiotic discovered by Y. Koyama, who derived it from *Paenibacillus polymyxa* in 1947 [5]. Colistin is effective against most Gram-negative bacteria but is ineffective against Gram-positive bacteria, anaerobic bacteria, and mycoplasmas [6]. It targets the lipopolysaccharide (LPS) of GNB out-membranes. The divalent cations, including Ca^2+^ and Mg^2+^, could be displaced by colistin from the anionic phosphate groups of LPS and destabilize the out-membrane, ultimately leading to the leakage of intracellular contents and bacterial death [7,8]. Early on, colistin was replaced in clinics by new and more effective antibiotics, due to reports of nephrotoxic and neurotoxic adverse events [9,10,11]. However, in the 1990s, with the emergence of MDR and XDR Gram-negative bacteria, colistin re-emerged as the last-resort treatment against superbugs [12]. Alarmingly, the plasmid-borne colistin-resistant *mcr* gene has spread in GNB worldwide, limiting the use of colistin and leaving clinicians with few choices among the existing antibiotics [13].

A large body of literature supports the hypothesis that a combination therapy of colistin may be a novel therapeutic option for multidrug-resistant bacteria [14,15]. For example, Ku et al. reported the synergistic effect of colistin in combination with fosfomycin in a mouse model of MDR *A. baumannii* pneumonia [16]. Amorolfine (AMO, Figure 1A) is a morpholine antifungal drug that inhibits Δ14-sterol reductase and cholestenol Δ-isomerase, which depletes ergosterol and causes ignosterol to accumulate in the fungal cytoplasmic cell membranes [17]. AMO is commonly applied as a lacquer for onychomycosis but is also used for superficial dermatomycosis [18,19]. However, there has been no report on the synergistic activity between AMO and colistin against *A. baumannii*. Here, we found that AMO could potentiate the sensitivity of colistin against *A. baumannii*, both in vitro and in vivo, providing a new and attractive combination therapy for *A. baumannii* infection in the future.

## 2. Results

### 2.1. In Vitro Synergistic Effect of the Amorolfine and Colistin Combination on Gram-Negative Bacteria

To determine the synergistic activity of AMO with colistin, the MIC of AMO was first determined (Figure 1B). Here, AMO is considered to have no antimicrobial activity against the strains tested in the table with an MIC value greater than 256 mg/L, regardless of whether it is being used against Gram-negative or Gram-positive bacteria, except for the outer membrane-deficient strain of *A. baumannii*. We then tested the synergistic activity of AMO in combination with colistin (Figure 1C–F). For four Gram-negative bacteria, using colistin in combination with AMO showed a 4- to 16-fold decrease in MIC values, including *A. baumannii* ATCC19606 (FICI = 0.094), *E. coli* ATCC 25922 (FICI = 0.28), *K. pneumoniae* ATCC13883 (FICI = 0.28) and *P. aeruginosa* ATCC9027 (FICI = 0.28). These results collectively suggest the general synergistic activity of AMO in combination with colistin against a broad spectrum of Gram-negative bacteria, with the best synergistic activity seen against *A. baumannii*.

### 2.2. Synergistic Effect of the Amorolfine and Colistin Combination on A. baumannii and Inhibition of Biofilm Formation

Against *A. baumannii*, a common clinical opportunistic pathogen and multidrug-resistant bacterium, the combination of amorolfine and colistin showed the best synergistic effect. To further assess the clinical value of the combination therapy, six clinically resistant *A. baumannii* strains were selected for testing, including two colistin-resistant strains. The combination showed good synergistic effects for all strains except the pmrA strain [20], which it was ineffective against due to the strain’s high resistance to colistin (Figure 2). In addition, for the pmrA strain, a decrease in bacterial concentration was observed in some of the wells of the 96-well plate test species that were co-administered, indicating that the synergistic effect was still present. Meanwhile, for colistin medium-resistance strain ab39, the combination still exhibited synergistic activity. These findings indicated that the synergistic activity of the combination of AMO and colistin remained unaffected by colistin resistance.

Time-kill experiments revealed that AMO monotherapy exhibited no bactericidal effect on *A. baumannii*, while colistin monotherapy led to a reduction in bacterial concentration during the initial phase. However, the bacteria continued to proliferate after 8 h. In contrast, the combination of AMO and colistin resulted in bacterial concentrations reaching the limit of detection (Figure 3A). The capacity of AMO to impede biofilm formation was evaluated using the CV staining method. The results of the checkerboard analysis demonstrated that AMO significantly inhibited *A. baumannii* biofilm formation by approximately 29% at 4 μg/mL, exhibiting a dose-dependent decrease (Figure 3B).

### 2.3. Effect of Amorolfine on the Membrane-Damaging Capacity of Colistin

A comparison was made of the synergistic effect of AMO in combination with antibiotics on different mechanisms (Figure 4). AMO was found to demonstrate synergistic antibacterial activity exclusively with colistin, which targets the outer membrane (OM). Consequently, the hypothesis was formulated that AMO might enhance OM disruption synergistically with colistin.

NPN was used to detect OM permeability by the enhancement of fluorescence upon binding to the hydrophobic part of the phospholipid bilayer [21]. The fluorescence intensity of NPN increased dramatically when AMO was combined with colistin (Figure 5A), indicating that the combination observably improved the permeability of the OM. Concurrently, both AMO and colistin have been observed to induce outer membrane disruption, a process that can be counteracted by excess Mg^2+^ (Figure 5B) [22]. This observation suggests the possibility of a shared mechanism for the binding of outer membrane cations by these two compounds. In order to gain further insight into the effect of colistin and AMO combination treatment on bacterial inner membrane (IM) permeability, PI was used as it binds to nucleic acids in membrane-damaged bacteria [23]. The combination treatment of AMO and colistin in this study resulted in a significant increase in IM permeability at low AMO concentrations (Figure 5C). Unexpectedly, the fluorescence values were instead reduced with high concentrations of AMO. Following the implementation of a fluorescence-based analysis, it was determined that AMO exhibits interference with PI binding to DNA, a phenomenon that is likely attributable to the capacity of AMO to bind DNA itself (Figure 5D).

### 2.4. Binding of Amorolfine to DNA Leads to Potential Bactericidal Activity

Despite the fact that AMO is not effective against Gram-negative or -positive bacteria, it has been demonstrated to possess bactericidal activity against outer membrane-deficient *A. baumannii* and shows synergistic activity with colistin, which suggests the potential for biological activity. In addition to the disruptive effect of AMO on the outer membrane, it is conceivable that the synergistic ability of AMO with colistin stems from its potential biological activity, in particular, its capacity to bind to DNA. In the absence of reports indicating AMO binding to DNA, further experimentation was conducted to investigate their binding mechanism.

Ultraviolet-visible absorption spectroscopy can be a common method to study DNA stability and its interaction with small ligand molecules. UV-vis analysis of CT-DNA in the absence and presence of AMO and the uptake of free AMO, CT-DNA, and its complex was conducted (Figure 6A). The results suggested a complex formation between AMO and DNA as the sum of the absorbance values of AMO and free CT-DNA was different from the complex. Moreover, with increasing concentrations of AMO, the absorption increased and hyperchromism was observed (Figure 6B). These results suggested a possible non-covalent interaction between AMO and CT-DNA.

Fluorescence spectroscopy is an effective method to study the DNA binding mode [24]. In general, EB and Hoechst 33258 bind to DNA through the insertion and groove binding modes, respectively, after which there is an increase in the emission spectra. If the drug binds to DNA in a similar mode, the dye will be displaced from the DNA helix, resulting in reduced fluorescence emissions [25]. Here, the emission spectra of the EB-DNA complex increased with increasing AMO concentration (Figure 7A), while the emission spectra of the Hoechst-DNA complex decreased (Figure 7B). These results showed that AMO binds to DNA primarily through a minor groove binding mode, but this binding leads to structural damage and base exposure of the DNA, which, in turn, binds better to the EB, leading to an increase in the emission spectrum of the EB-DNA complex.

We performed molecular docking studies using the AutoDock Vina docking modules. The molecular docking result showed that the binding mode of AMO is groove-binding in nature, which is consistent with previous experimental results (Figure 8). The maximum affinity conformation of AMO with DNA showed that AMO interacts with DNA near the AT region and had a minimum binding energy of −6.0 kcal/mole, which is similar to groove-binding drugs that prefer to interact with AT-rich regions rather than GC-rich regions [26,27].

To further evaluate the synergistic antibacterial mechanism of AMO and colistin [28], the structure and morphology of bacteria treated with different drugs were observed using TEM. Compared with the control group, bacterial structure was dramatically affected in both the colistin and AMO monotherapy groups (Figure 9). However, the effect of colistin on the bacteria was mainly focused on the structure of the envelope, giving it the appearance of discontinuous or wrinkled membranes, as shown by the red arrows, which finding is consistent with its mechanism targeting the outer membrane. Meanwhile, in the AMO-treated group, the cytoplasm within the bacteria manifested aggregation, as shown by the red circle. Previous reports have indicated that DNA-binding drugs induce a comparable phenomenon, resulting in the concentration of cytoplasm in the nucleoid region of the bacterial cells [29,30]. In contrast, the combined group exhibited both phenomena simultaneously, thereby suggesting that the DNA binding of AMO and the envelope disruption of colistin occurred concurrently. This finding may provide a rationale for the observed synergy between AMO and colistin.

### 2.5. Amorolfine Enhances the Efficacy of Colistin In Vivo Infection Models

Given that the combination of AMO and colistin showed outstanding synergistic bactericidal activity against *A. baumannii* in vitro, we next evaluated its therapeutic efficacy in vivo with a mouse sepsis model. First, the hemolytic activity of AMO was assessed using mouse blood erythrocytes to ensure appropriate and safe drug dosing. The in vivo drug concentration in AMO was set at 16 mg/kg, as hemolysis is generally considered to be in the safe range of less than 5% (Figure 10A) [31]. In the mouse sepsis model, the survival rate was used to evaluate the effect of synergistic therapy. All mice treated with saline or AMO died within 24 h, while colistin monotherapy slightly prolonged the survival time of the mice, but all still died within 36 h. In comparison, the combination of 0.5 mg/kg colistin and 16 mg/kg AMO increased the survival rate to 37.5% (3/8) (Figure 10B). This result demonstrated the significant synergistic antibacterial effects of the combination of colistin and AMO in the treatment of *A. baumannii* infectious in vivo, highlighting its potential for clinical application.

## 3. Discussion

In 2019 alone, antimicrobial resistance was estimated to have directly resulted in more than 1.2 million deaths, and, without preventative measures, this will increase to approximately 10 million by 2050 [32,33]. This fact has emerged as a pressing concern within the domain of public health. However, the development of new antibiotics faces numerous challenges, as evidenced by the fact that only 43 phase 1–3 clinical trials were registered for antibiotics by the end of 2020, compared to 1300 trials for anticancer agents [34]. Antibiotic synergy is a promising strategy to combat antibiotic-resistant bacteria, particularly those antibiotics that have become neglected and disused because of toxicity or moderate activity [35,36,37].

In this study, we identified a new potential colistin adjuvant—amorolfine, which enhanced the bactericidal activity of colistin against *A. baumannii*, both in vitro and in vivo. Amorolfine is a morpholine antifungal drug for topical use, like toenail onychomycosis [38], but it has never been reported that AMO has potential antibacterial activity. The results of the checkerboard studies and time-killing curve assay showed that a combination of AMO and colistin dramatically increased the bactericidal activity of colistin against *A. baumannii*, compared to monotherapy. In addition, the combination had a synergistic bacteriostatic effect on all clinical strains tested, regardless of whether they were colistin-resistant or not. The results of the biofilm formation inhibition assay demonstrate that AMO exhibits anti-biofilm activity. Biofilm is a three-dimensional bacterial community structure composed of polysaccharides, proteins, extracellular DNA, and others [39,40]. It represents an important barrier for bacteria to resist the harsh external environment, especially disinfectants and antibiotics in hospital environments [41]. In previous experiments, we demonstrated the DNA-binding capability of AMO, which may lead to the absence of eDNA during biofilm formation and to its inhibition.

Herein, in order to explore how AMO and colistin work synergistically, a comparison was made of other types of antibiotics with different mechanisms in combination with AMO. The results demonstrated that AMO is only synergistic with colistin, suggesting that the synergy between the two is dependent on colistin’s bactericidal mechanism. In consideration of the mechanism by which colistin targets LPS, resulting in bacterial OM disruption [42], bacterial OM permeability was measured using NPN following the combination treatment of AMO and colistin. This demonstrated that AMO enhanced the OM breaking efficiency of colistin; it was influenced by Mg^2^, which is similar to colistin. In accordance with the theory proposed by Buchholz et al. [43], it is hypothesized here that the similar outer membrane disruption mechanism of AMO and colistin contributes to the initial disruption of the outer membrane structure, thereby facilitating the release of additional LPS binding sites for binding to colistin. Membrane permeability is one of the most important factors for antibiotic activity, and membrane-targeted antibiotics, like colistin, usually show rapid bactericidal activity. The combination of AMO and colistin showed a rapid bactericidal effect in the time-killing curve; the bacterial count dropped rapidly within 4 h. Furthermore, the PI measurement of IM permeability reflected the synergistic enhancement of IM damage by AMO and colistin. Concurrently, we found that AMO seems to affect the fluorescence intensity of the nucleic acid fluorescent dye PI, suggesting that AMO may be a DNA-binding drug.

The ultraviolet-visible spectroscopy and competitive displacement assays showed that AMO binds to DNA and displayed a minor groove binding mode [44]. DNA is an important drug target [45]; for example, the anticancer drug cisplatin exerts its inhibitory effect on DNA replication and cell growth by binding to the DNA [46]. Similarly, quinolone antibiotics are known to bind to DNA and DNA topoisomerases, thereby facilitating their bactericidal activity [47,48]. We used molecular docking to investigate the binding modes of AMO and DNA; consistent with previous experimental findings, we found that AMO binds to DNA via a minor groove binding mode. The DNA-binding capacity of AMO and the membrane-disrupting activity of colistin in bacterial structures were further confirmed by TEM images.

Finally, we tested the synergistic effect of AMO and colistin in a mouse *A. baumannii* sepsis model, and the combination therapy was effective in improving survival in mice. In consideration of the nephrotoxicity and neurotoxicity of colistin and the prevalence of colistin-resistant *A. baumannii*, its combination with AMO has the potential to reduce the actual drug dosage while ensuring the efficacy of colistin [49]. This combination represents a viable method to overcome the limitations of the clinical use of colistin.

## 4. Materials and Methods

### 4.1. Reagents and Bacterial Strains

AMO (CAS No. 78613-38-4) and colistin sulfate (CAS No. 8068-28-8) were purchased from Macklin Biochemical Co., Ltd. (Shanghai, China). CT-DNA (CAS No. 91080-16-9) was purchased from Meryer Co., Ltd. (Shanghai, China), and porcine mucin (CAS No. 84082-64-4) was purchased from Shanghai Yuanye Bio-Technology Co., Ltd. (Shanghai, China). All other antibiotics were obtained from Sangon Biotech Co., Ltd. (Shanghai, China). All strains used in this study were isolated and preserved in our laboratory.

### 4.2. The Antimicrobial Susceptibility Test and Checkerboard Assays

The minimum inhibitory concentration (MIC) values of antibiotics were determined by the broth microdilution method according to the Clinical and Laboratory Standards Institute (CLSI) [50]. Bacteria were cultured to the logarithmic growth stage and the cell density was adjusted to 10^6^ CFU. Antibiotics were 2-fold serial dilutions in the Mueller–Hinton (MH) culture. The MIC was recorded with no visible growth after incubation at 37 °C for 16 h. Synergistic antibacterial activity was determined by calculating the fractional inhibitory concentration index (FICI) through the checkerboard experiment. FICI ≤ 0.5 was defined as synergistic, 0.5 ≤ FICI ≤ 1 was defined as addition, 1 < FICI ≤ 4 was defined as indifference, and FICI > 4 was defined as antagonism.

### 4.3. The Time-Kill Assay

The time-kill assay was conducted to evaluate the synergistic effect of the tested combination, following the results obtained from the checkerboard assay. Briefly, *A. baumannii* ATCC19606 was inoculated into 5 mL of MH containing colistin (1 μg/mL) and AMO (16 μg/mL), either alone or in combination, at a concentration of 10^7^ CFU/mL. The various drug-treated groups were subjected to cultivation at 37 °C with shaking at 220 rpm. At different time points (0, 1, 2, 4, 8, and 24 h), appropriate dilutions were made, and colony-forming units (CFU) were enumerated on MH agar plates.

### 4.4. The Biofilm Formation Assay

The biofilm inhibition assay was conducted on 96-well flat-bottom microtiter plates [51]. *A. baumannii* ATCC19606 was inoculated into 100 μL MH containing AMO (0, 4, 16, and 64 μg/mL) at a cell density of 10^6^ CFU/mL. Six wells were used for each agent concentration. After incubating the 96-well plate at 37 °C for 24 h, planktonic cells were discarded, and the wells were washed twice with PBS. Subsequently, the plates were air-dried to facilitate the straightforward fixation of the biofilm. Staining and destaining were carried out using 1% crystal violet solution and ethanol, respectively. Between these steps, the wells were washed with 1 × PBS and left to air-dry. Finally, the absorbance of each well was measured at 595 nm using a microplate reader.

### 4.5. Membrane Permeability Evaluation

*A. baumannii* ATCC19606 was grown overnight at 37 °C with shaking at 220 rpm. Then, the cultures were washed and suspended with 5 mM HEPES (pH 7.0, plus 5 mM glucose). In the same buffer, the OD_600_ of the bacterial suspension was standardized to 0.5 and the fluorescent dye was added. After incubation at 37 °C for 30 min, the bacterial suspension was mixed with AMO and colistin, either alone or in combination. After incubation for 1 h, 200 µL of bacterial suspension was added to the 96-well plate. Subsequently, fluorescence intensity was measured by an Infinite 200 PRO plate reader (Tecan, Männedorf, Switzerland). A fluorescent probe of 1-N-phenylnaphthylamine (NPN) (10 μM) was used to evaluate the outer membrane (OM) integrity, using an excitation wavelength of 350 nm and an emission wavelength of 420 nm, and propidium iodide (PI) (5 μM) was used to evaluate the inner membrane (IM) integrity using an excitation wavelength at 535 nm and an emission wavelength of 615 nm.

### 4.6. Ultraviolet-Visible Spectroscopy

The interaction of AMO with CT-DNA was studied using UV-vis spectroscopy (Shimadzu, Kyoto, Japan). A fixed concentration of CT-DNA (40 μM) was titrated with varying AMO concentrations from 0 to 120 μM in 10 mM Tris-HCl (pH 7.2) buffer. To obtain the absorption spectra, the required amount of AMO was added to both the compound solution and the reference solution to eliminate the absorbance of the AMO itself. The samples were incubated at 25 °C for 24 h and the spectra were scanned from 240 to 300 nm.

### 4.7. The Competitive Displacement Assays

Ethidium bromide (EB) and acridine Hoechst 33258 are intercalating dyes and the minor groove binding dye is separate; they are commonly used to study the binding mode of drugs with CT-DNA. A fixed concentration (5 μM) of EB or Hoechst 33258 was taken in the presence of 50 μM CT-DNA in 10 mM Tris-HCl (pH 7.2) buffer. The EB-DNA complex was excited at 471 nm and emissions spectra were recorded from 550 to 750 nm in the presence of an increasing concentration of AMO. Similarly, the Hoechst-DNA complex was excited at 343 nm, and emission spectra were recorded from 400 to 500 nm.

### 4.8. The Molecular Docking Assay

The CT-DNA sequence d(CGTGAATTCACG)2 dodecamer (PDB ID: 5T4W) was obtained from the Protein Data Bank. Receptor (DNA) and ligand (complex) files were prepared using MGLtools1.4.2 and OpenObabel3.0.0. A molecular docking program, AutoDock Vina v1.2.5, was used to study the interaction of AMO with DNA. Docking pockets were predicted using CavityPlus v1.0 and the first ranked result was used as the docking center. The grid box was set as a square with 18 Å sides, for which 20 conformations were output, and the optimal conformations were selected according to docking scores. PyMol 3.1.3.1 (Educational Version) was used to visualize the docked complex.

### 4.9. Transmission Electron Microscopy (TEM)

The morphological appearance and morphometric analysis of the cell membrane of *A. baumannii* were determined using TEM. Briefly, the bacterial suspensions were washed twice with PBS and exposed to 32 μg/mL AMO, 1 μg/mL colistin alone, or their combination (32 μg/mL AMO + 1 μg/mL colistin) at 37 °C. After 4 h, the cells were centrifuged, collected, and fixed using 2.5% glutaraldehyde at 4 °C overnight. Then, the samples were fixed with 1% osmium acid, embedded in Epon 812 epoxy resin, sectioned, stained with 2% uranyl acetate, and observed using TEM (HT7800 HITACHI, Tokyo, Japan).

### 4.10. The Hemolysis Test

To further evaluate the security of AMO, a mouse red blood cell (RBS) hemolysis assay was used. Various concentrations of AMO from 0 to 128 μg/mL were incubated with the RBC suspension at 37 °C for 2 h. Meanwhile, 0.1% Triton X-100 served as the corresponding positive control group. Following incubation, the supernatant was obtained by centrifugation at 12,000 rpm for 5 min, and its absorbance at 545 nm was measured. Hemolysis rate (%) = (OD_545_ experimental group − OD_545_ negative control group)/(OD_545_ positive control group − OD_545_ negative control group).

### 4.11. The Mouse Infection Model In Vivo

Specified pathogen-free (SPF) female BALB/c mice (5 to 6 weeks old, ~20 g) were randomly grouped, with eight mice in each group. Before the formal experiment, the mice were fed for one week to adapt to the environment and then divided equally into four groups based on weight (*n* = 8 per group). Referring to the sepsis model established by Harris [52], minor modifications were made based on our research. Briefly, *A. baumannii* ATCC19606, cultured under exponential growth conditions, was collected and the bacterial concentration was adjusted to OD_600_ = 1.0 with saline. The inoculum was prepared by mixing the bacterial solution with 10% porcine mucin solution at a ratio of 1:1. Mice were injected intraperitoneally with 0.5 mL of the inocula. Then, after 1 h, the mice were injected with saline, AMO (16 mg/kg), colistin (0.5 mg/kg), and their combination (0.5 mg/kg + 16 mg/kg). The mice were observed, and death was recorded within 72 h.

### 4.12. Statistical Analysis

The sample size for each statistical analysis was greater than or equal to three. All data were analyzed using GraphPad Prism 9.0 software. Comparisons between two groups were calculated using two-way ANOVA (*, *p* < 0.05; **, *p* < 0.01; ***, *p* < 0.001; ****, *p* < 0.0001).

## 5. Conclusions

In conclusion, our study suggests the synergistic activity of AMO and colistin against GNB, notably *A. baumannii*; combination therapy enhanced the therapeutic efficacy of colistin in a mouse sepsis model. Moreover, mechanistic studies suggested that AMO enhanced the ability of colistin to damage the membrane and that AMO may cross the OM via this process to bind to intracellular DNA and exert additional bactericidal activity. The dual-target bactericidal mechanism of AMO–colistin will effectively reduce the possibility of bacterial drug resistance, which finding is of high clinical value [53]. In addition, we are currently engaged in the modification of AMO, with the objective of producing molecules that exhibit enhanced antimicrobial activity and identifying additional application scenarios.

## Figures and Tables

**Figure 1 ijms-26-03312-f001:**
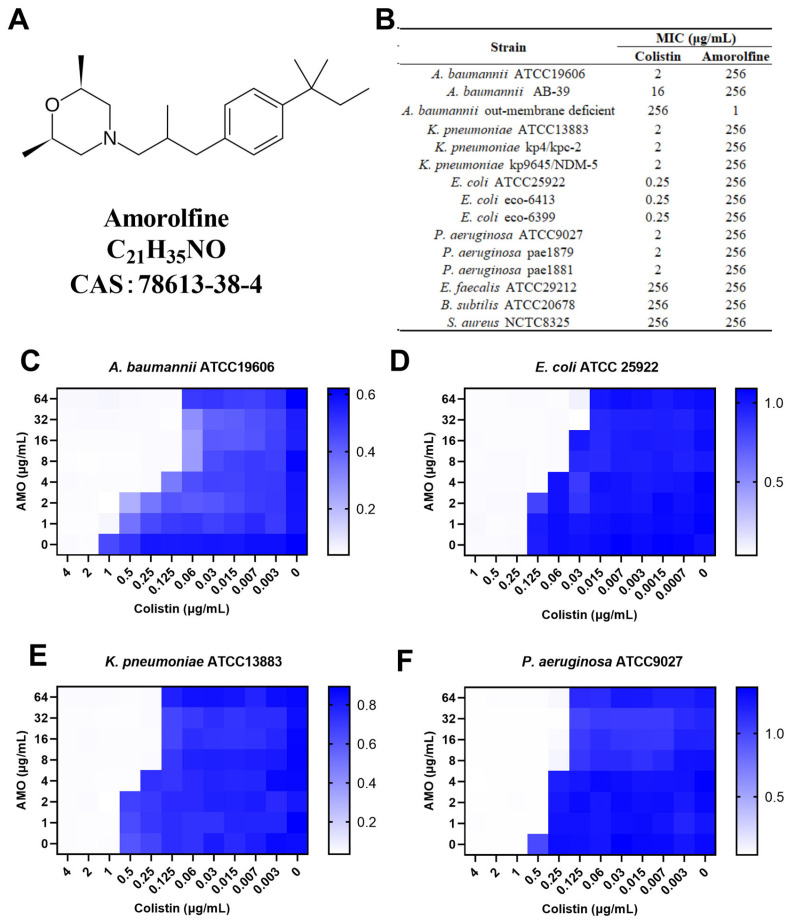
(**A**) The chemical structure of amorolfine. (**B**) The MIC values of AMO and colistin. (**C**–**F**) The synergistic effect of AMO combined with colistin against GNB, as shown by the checkerboard assay. The data represent the values at OD_600_ nm of bacterial culture. Dark blue regions represent higher cell density.

**Figure 2 ijms-26-03312-f002:**
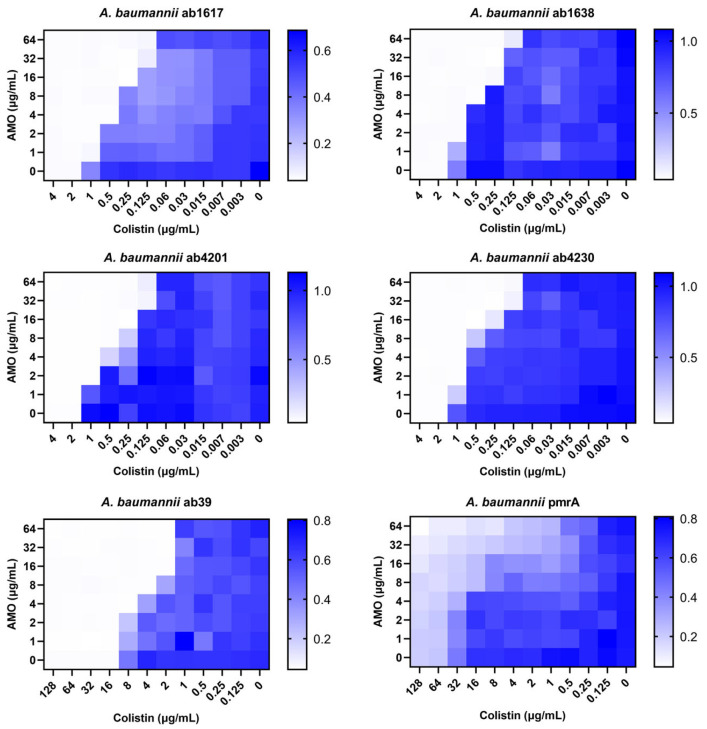
The synergistic effect of AMO combined with colistin against clinical *A. baumannii* isolates (ab1617, ab1638, ab4230-MEM, CN, AMP; ab4201-MEM, CN; ab39, pmrA-COL), as shown by the checkerboard assay. MEM—MeropeneM; CN—Cefalexin; AMP—Ampicillin; COL—Colistin.

**Figure 3 ijms-26-03312-f003:**
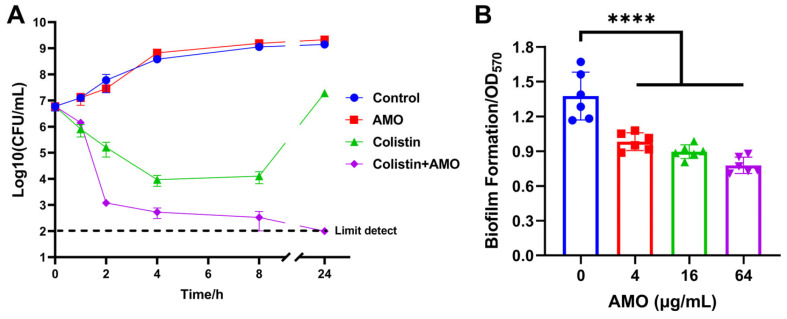
(**A**) Time–kill curves of *A. baumannii* ATCC19606 incubated with colistin (1 μg/mL), AMO (16 μg/mL) alone, or their combination. (**B**) The biofilm formation of *A. baumannii* ATCC19606 after exposure to different concentrations of AMO. (****, *p* < 0.0001).

**Figure 4 ijms-26-03312-f004:**
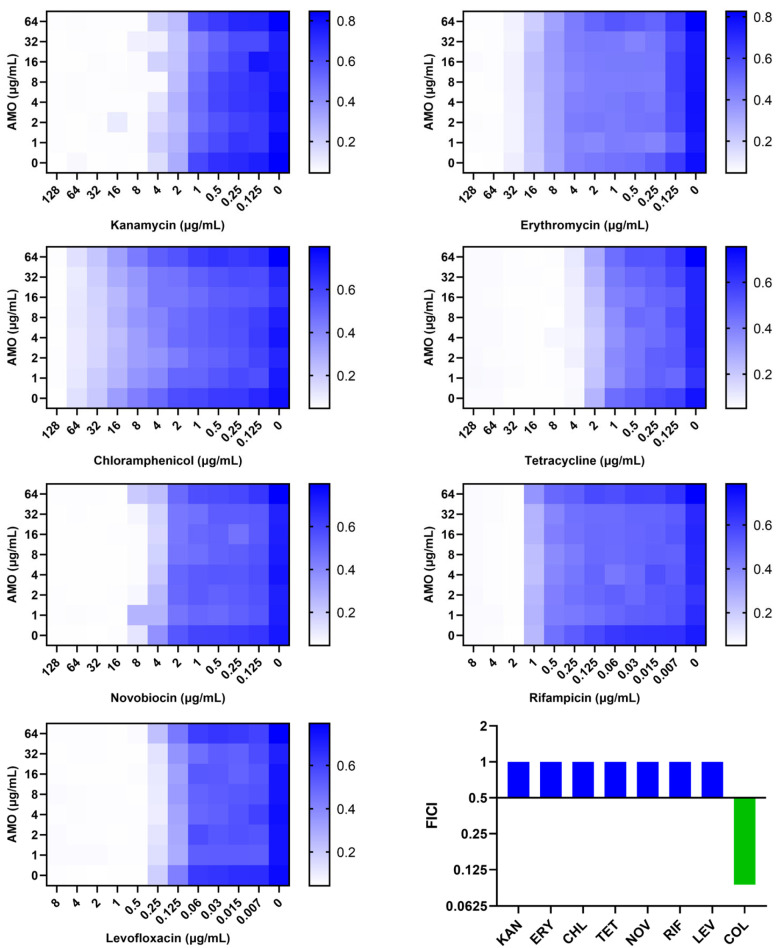
The synergistic effect of AMO combined with different antibiotics against *A. baumannii* ATCC19606, as shown by the checkerboard assay, and their FICI.

**Figure 5 ijms-26-03312-f005:**
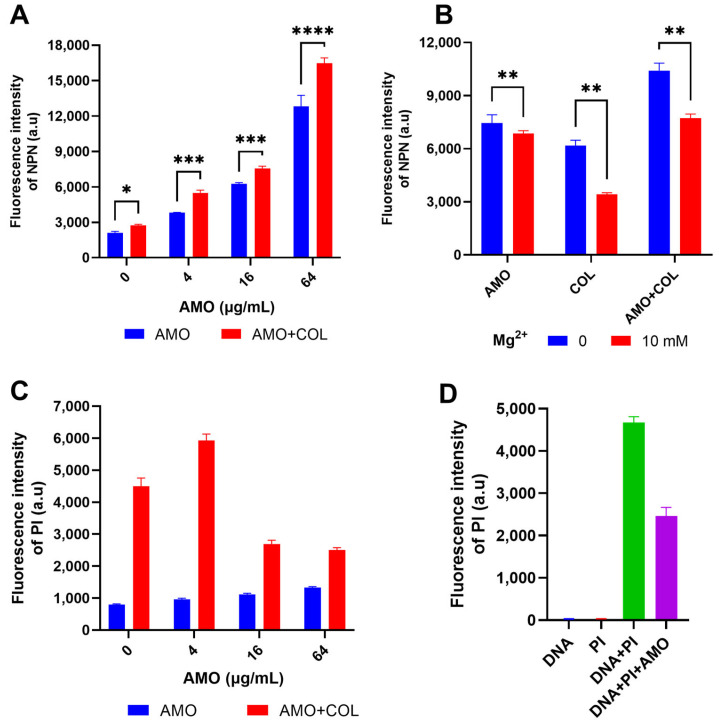
(**A**) Detection of the OM permeability of *A. baumannii* ATCC19606 under different concentrations of AMO (0, 4, 16 and 64 μg/mL), with or without colistin (1 μg/mL). (**B**) Detection of the OM permeability of *A. baumannii* ATCC19606 treated with AMO (16 μg/mL), colistin (1 μg/mL) alone, or their combination, with or without Mg^2+^. (**C**) Detection of the IM permeability of *A. baumannii* ATCC19606 under different concentrations of AMO (0, 4, 16, and 64 μg/mL) with or without colistin (1 μg/mL). (**D**) The fluorescence values for the different systems. (*, *p* < 0.05; **, *p* < 0.01; ***, *p* < 0.001; ****, *p* < 0.0001).

**Figure 6 ijms-26-03312-f006:**
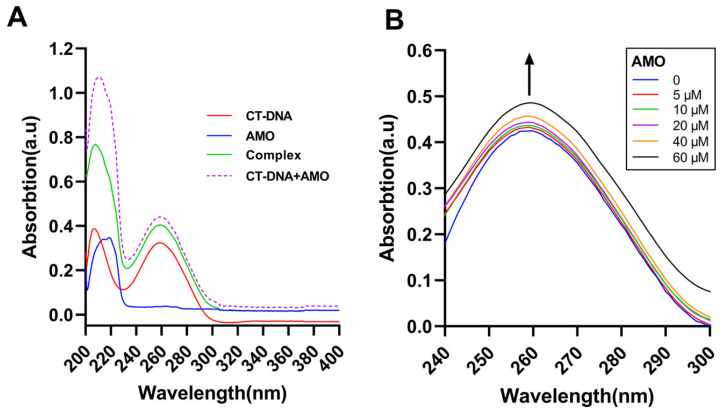
(**A**) UV-vis absorption spectra of CT-DNA (30 μM), AMO (2 μM) alone, and their complex. (**B**) UV-vis absorption spectra of CT-DNA (40 μM) in the presence of increasing concentrations of AMO (0–60 μM). The arrow shows the emission intensity changes upon increasing AMO concentrations.

**Figure 7 ijms-26-03312-f007:**
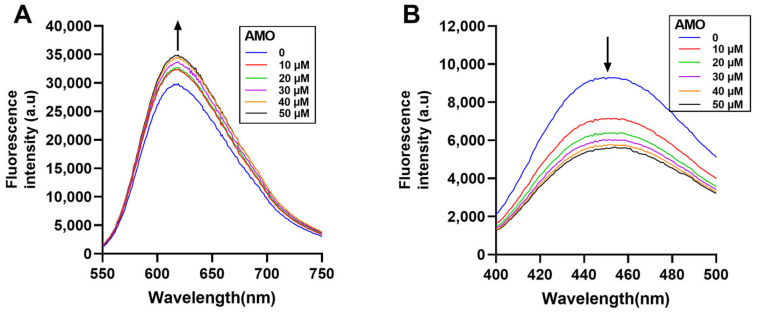
Displacement assays using nucleic acid fluorescent dyes. The CT-DNA and EB (**A**) complex and CT-DNA and Hoechst 33258 (**B**) complex were excited in the presence of increasing concentrations of AMO (0–50 μM). The arrow shows the emission intensity changes upon increasing AMO concentrations.

**Figure 8 ijms-26-03312-f008:**
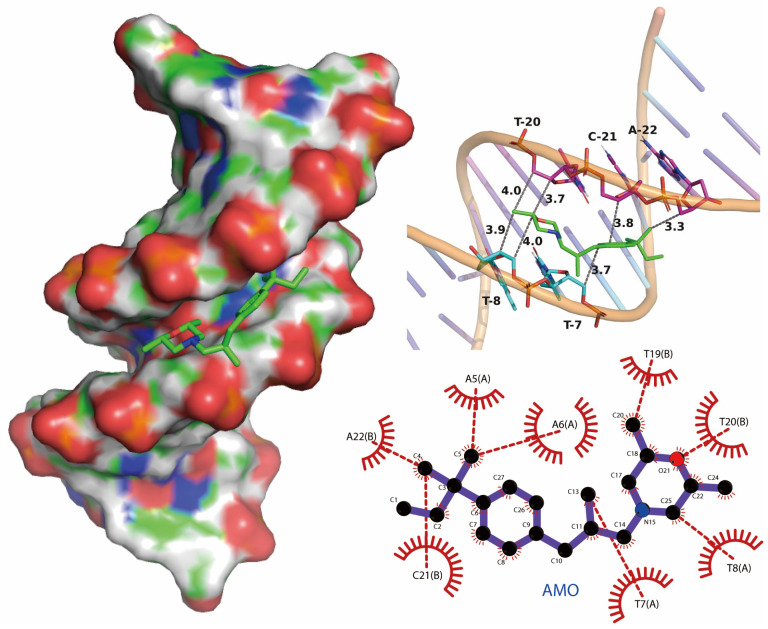
Potential binding mode between AMO and DNA using molecular docking.

**Figure 9 ijms-26-03312-f009:**
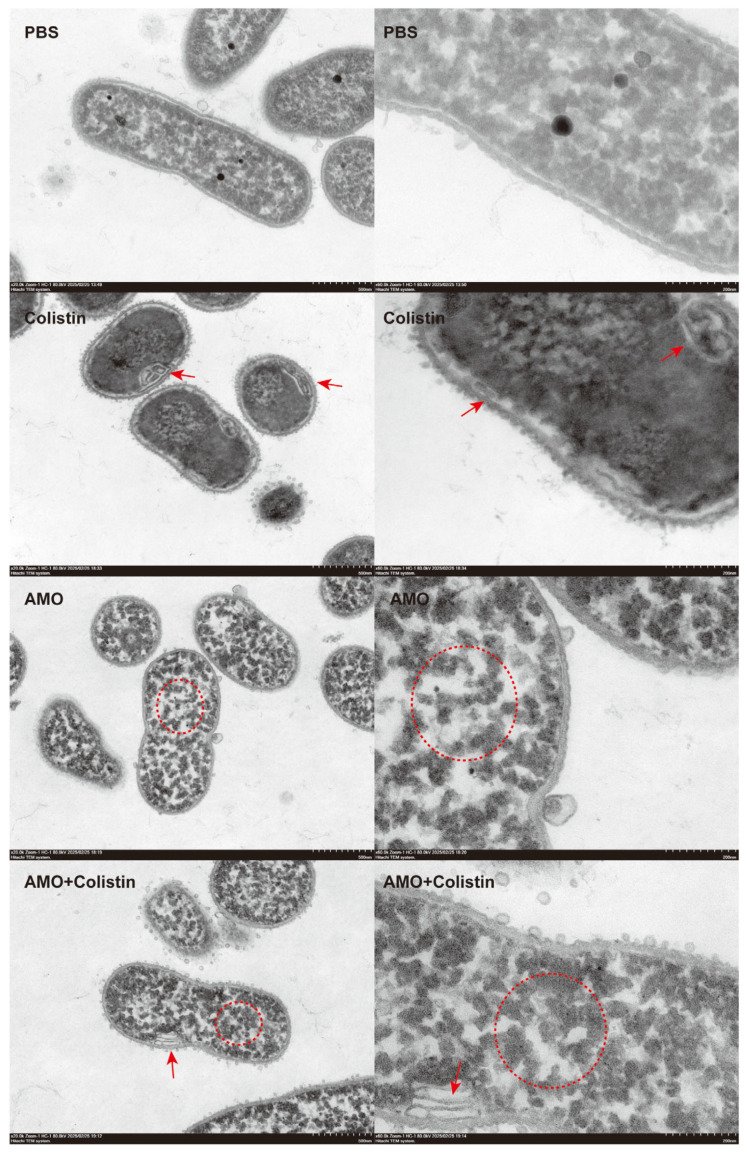
Morphological analysis of *A. baumannii* ATCC19606 treated with colistin (1 μg/mL), AMO (32 μg/mL) alone, or their combination, elucidated by transmission electron microscopy (scale bar: 500 nm and 200 nm).

**Figure 10 ijms-26-03312-f010:**
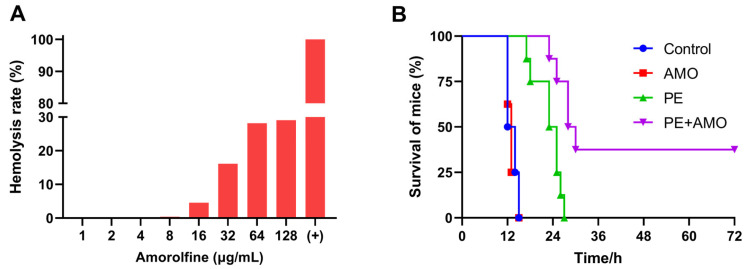
(**A**) Hemolysis rate of AMO. (**B**) The survival rates of mice (*n* = 8 per group) in the mouse *A. baumannii* ATCC19606 sepsis model.

## Data Availability

The original contributions presented in this study are included in the article. Further inquiries can be directed to the corresponding author.

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
