# Peer review of "Synergistic Antibacterial Activity of Amorolfine Combined with Colistin Against Acinetobacter baumannii"

_ijms, 2025, doi:10.3390/ijms26073312_

Round 1
Reviewer 1 Report
Comments and Suggestions for Authors
Minor comments:
(Line 26-30) How does the fractional inhibitory concentration index (FICI) of AMO-colistin compare with other antibiotic combinations against A. baumannii?
(Line 45-50) What are the molecular mechanisms behind AMO’s ability to enhance colistin’s membrane-disrupting effects?
(Line 73-76) Could AMO’s DNA-binding properties contribute to bacterial resistance in the long term?
(Line 103-107) How does AMO interact with A. baumannii’s lipopolysaccharides compared to other colistin adjuvants?
(Line 138-150) Would different structural modifications of AMO improve its synergistic effect with colistin?
(Line 175-183) How does AMO-colistin synergy vary among different clinical isolates of A. baumannii, particularly those with diverse resistance mechanisms?
(Line 218-228) How does the AMO-colistin combination affect bacterial biofilm formation over prolonged treatment periods?
(Line 323-328) What are the potential off-target effects of AMO in mammalian cells, and how can they be mitigated?
(Line 364-374) How does AMO affect colistin's pharmacokinetics, particularly in terms of tissue distribution and clearance?
(Line 417-421) Could AMO be structurally modified to enhance its bacterial uptake without increasing toxicity?
(Line 464-470) What alternative experimental models (e.g., organoid models or 3D tissue cultures) could be used to validate the in vivo findings?
(Line 498-502) How does the AMO-colistin combination perform in polymicrobial infections where A. baumannii is present alongside other pathogens?
Author Response
Dear Reviewer:
Thank you for your valuable time and insightful comments on our manuscript. We sincerely appreciate the constructive feedback provided, which has significantly improved the clarity and rigor of this work. All changes have been highlighted in the revised manuscript, and detailed point-by-point responses to each comment are provided following:
Comments 1: (Line 26-30) How does the fractional inhibitory concentration index (FICI) of AMO-colistin compare with other antibiotic combinations against A. baumannii?
Response 1: A lot of polymyxin-drug combinations were previously tested, a number of which were reported compounds, for example, fusidic acid-colistin[1] (FICI=0.003). Nonetheless, the utilisation of AMO as an adjuvant was unprecedented. Furthermore, because of issues such as strain heterogeneity, the span of FICI for the same drug combination can be quite large, similar to this literature[2], where the FICI for netilmicin-imipenem combination ranged from 0.25-1.026.
Comments 2: (Line 45-50) What are the molecular mechanisms behind AMO’s ability to enhance colistin’s membrane-disrupting effects?
Response 2: Thank you for pointing this out and we will add the following content to the discussion section. Here, current studies on the mechanism of outer-membrane disruption by colistin suggest that cations stabilising the LPS membrane structure are replaced by colistin, and leading to structural disruption. Our study demonstrates that AMO possesses a specific outer membrane disruption function and is influenced by cations (Mg2+). It is hypothesised that AMO employs a comparable molecular mechanism to that of colistin. But how do the similar molecular mechanisms of the two synergize? Here, we cite a theory from the literature to explain it [3]. This literature suggests that LPS is initially limited in the number of sites used to bind to colistin, however, as the structure is disrupted, more binding sites will be exposed and bind to the colistin, contributing to further structural disruption. We speculate that AMO may have initially facilitated the exposure of the LPS binding site, but further validation of this conclusion is still required
Comments 3: Line 73-76) Could AMO’s DNA-binding properties contribute to bacterial resistance in the long term?
Response 3: This is an issue which requires further elucidation. We must recognize that resistance is almost inevitable, but because of the following three reasons we believe that the possibility of resistance can be minimised, 1) AMO itself is ineffective against bacteria, and its use as a single agent does not constitute a stressful screening; 2) AMO's DNA-targeting has a relatively low success rate of resistance mutations compared to the protein-targeting of conventional antibiotics, but it may be possible to achieve this through drug efflux mechanisms; 3) AMO-colistin combination is based on a different mechanism of action, which increases the difficulty of resistance generation.
Comments 4: (Line 103-107) How does AMO interact with A. baumannii’s lipopolysaccharides compared to other colistin adjuvants?
Response 4: Current researches have focused on finding effective colistin-adjuvant combinations, with relatively few studies specifically on synergistic mechanisms. In addition to the speculation mentioned in comment 2, we are also concerned about the activation of the LPS transporter enzyme in colistin-novobiocin combinations to enhance activity of colistin [4], but the current evidence for AMO in this study focuses mainly on competition for Mg2+ to disrupt the structure of LPS.
Comments 5: (Line 138-150) Would different structural modifications of AMO improve its synergistic effect with colistin?
Response 5: It's exactly what you think, our next work plan includes structural modification of AMO to obtain compounds with stronger potentiating properties, and based on preliminary results our answer is that structural modification does affect its activity, for better or for worse.
Comments 6: (Line 175-183) How does AMO-colistin synergy vary among different clinical isolates of A. baumannii, particularly those with diverse resistance mechanisms?
Response 6: Thank you for pointing this out, we will label the clinical bacteria within the article for resistance. In fact, 4 of the 6 clinical bacteria are Meropene-resistant and do not involve the antimicrobial mechanism of polymyxin and AMO, so are sensitive to both. The remaining 2 strains are colistin-resistant but only affect colistin and not AMO (DNA-binding), so synergism is still evident despite the high MIC to colistin.
Comments 7: (Line 218-228) How does the AMO-colistin combination affect bacterial biofilm formation over prolonged treatment periods?
Response 7: We agree with your comment and we will add the following to the article. eDNA is one of the components of biofilms, and the DNA-binding ability of AMO may inhibit initial biofilm formation. Furthermore, we are trying to establish a galleria mellonella model to mimic the process to confirm whether the biofilm inhibitory activity of AMO can ameliorate this.
Comments 8: (Line 323-328) What are the potential off-target effects of AMO in mammalian cells, and how can they be mitigated?
Response 8: The antifungal target (Δ14-sterol reductase and cholestenol Δ-isomerase) of the AMO is not present in animal cells. However, its DNA binding mechanism may have the potential to result in toxicity. We are trying to improve drug targeting to bacteria rather than animal cells using liposomes, for example.
Comments 9: (Line 364-374) How does AMO affect colistin's pharmacokinetics, particularly in terms of tissue distribution and clearance?
Response 9: This is a pivotal issue, and one of the constraints on the in vivo use of drug combinations is the pharmacokinetic differences between the different drugs. However, the studies related to AMO illustrate that it is generally used for topical therapies such as skin, so there are almost no studies on pharmacokinetics. And the impact of AMO on the pharmacokinetics of colistin cannot be judged easily. But such works may be added when the drug has been structurally modified and formulated in future.
Comments 10: (Line 417-421) Could AMO be structurally modified to enhance its bacterial uptake without increasing toxicity?
Response 10: Our current results on structural modifications suggest that there are indeed cases where increased activity can be accompanied by reduced toxicity as far as hemolysis is concerned, but the magnitude is relatively limited, and we believe that formulation studies are likely to be more efficient.
Comments 11:(Line 464-470) What alternative experimental models (e.g., organoid models or 3D tissue cultures) could be used to validate the in vivo findings?
Response 11: We are currently establishing a galleria mellonella model for bulk in vivo screening of AMO activity after structural modification at a later stage, which is easier to operate and less costly than the mouse model.
Comments 12: (Line 498-502) How does the AMO-colistin combination perform in polymicrobial infections where A. baumannii is present alongside other pathogens?
Response 12: We agree with your idea that based on the complexity of the clinical infection situation, it is actually very worthwhile to study the content, but also very difficult. AMO has antifungal activity, and it is initially estimated that the AMO-colistin combination may be effective in the situation of fungal-Gram-negative bacteria co-infection. But the co-infection situation is quite complex and involves immune system, multiple microbes, drugs and its interactions. And it did produce a certain bactericidal effect in vitro mixed fungi-bacteria culture, but it is still difficult to establish the in vivo co-infection model.
Once again, we deeply appreciate your expertise and support in advancing this research.
- Phee, L. M.; Betts, J. W.; Bharathan, B.; Wareham, D. W., Colistin and Fusidic Acid, a Novel Potent Synergistic Combination for Treatment of Multidrug-Resistant Acinetobacter baumannii Infections. Antimicrob Agents Chemother 2015, 59, (8), 4544-50.
- Mackay, M. L.; Milne, K.; Gould, I. M., Comparison of methods for assessing synergic antibiotic interactions. Int J Antimicrob Agents 2000, 15, (2), 125-9.
- Buchholz, K. R.; Reichelt, M.; Johnson, M. C.; Robinson, S. J.; Smith, P. A.; Rutherford, S. T.; Quinn, J. G., Potent activity of polymyxin B is associated with long-lived super-stoichiometric accumulation mediated by weak-affinity binding to lipid A. Nature Communications 2024, 15, (1), 4733.
- Mandler, M. D.; Baidin, V.; Lee, J.; Pahil, K. S.; Owens, T. W.; Kahne, D., Novobiocin Enhances Polymyxin Activity by Stimulating Lipopolysaccharide Transport. J Am Chem Soc 2018, 140, (22), 6749-6753.
Reviewer 2 Report
Comments and Suggestions for Authors
Dear Authors
A very important topic, especially when we are looking for new antibacterial compounds, including those that enhance the activity of antibiotics against alarm bacteria. To date, there are no studies on the synergy of amorolfine and colistin against strains of alarm bacteria. The manuscript provides an opportunity to expand this knowledge
In my opinion, the topic is very important, a very good research concept, well-chosen and described research methodology. Well presented and interpreted results. Correctly drawn conclusions consistent with the presented evidence and arguments. Well written discussion. The work brings a lot of information that has practical significance and can be used in the treatment of infections caused by A. baumaii and not only.
A minor editorial note - improving references.
Best regards
Author Response
Dear Reviewer:
Thank you for recognizing this study and your review efforts. After gathering feedback from all parties, we have made the following modifications include:
- The dual-target bactericidal mechanism of AMO-colistin will effectively reduce the possibility of bacterial drug resistance, which is of high clinical value[1].
- It is hypothesised that AMO may inhibit biofilm formation by binding to the eDNA of the biofilm.
- In accordance with theory of Buchholz et al[2], it is hypothesised that the similar outer membrane disruption mechanism of AMO and colistin contributes to the initial disruption of the outer membrane structure, thereby facilitating the release of additional LPS binding sites for binding to colistin.
- Some details, labeling changes in manuscript.
All changes have been highlighted in the revised manuscript, we remain open to further revisions if needed. Once again, we deeply appreciate the reviewers' expertise in advancing this research.
- Martin, J. K.; Sheehan, J. P.; Bratton, B. P.; Moore, G. M.; Mateus, A.; Li, S. H. J.; Kim, H.; Rabinowitz, J. D.; Typas, A.; Savitski, M. M.; Wilson, M. Z.; Gitai, Z., A Dual-Mechanism Antibiotic Kills Gram-Negative Bacteria and Avoids Drug Resistance. Cell 2020, 181, (7), 1518-+.
- Buchholz, K. R.; Reichelt, M.; Johnson, M. C.; Robinson, S. J.; Smith, P. A.; Rutherford, S. T.; Quinn, J. G., Potent activity of polymyxin B is associated with long-lived super-stoichiometric accumulation mediated by weak-affinity binding to lipid A. Nature Communications 2024, 15, (1), 4733.